

# A trajectory data compression algorithm based on spatio-temporal characteristics

Yanling Zhong[1], Jinling Kong[1], Juqing Zhang[1], Yizhu Jiang[2], Xiao Fan[1] and Zhuoyue Wang[2]

[1] School of Geological Engineering and Geomatics, Chang'an University, Xi'an, Shaanxi, China
[2] School of Earth Science and Resources, Chang'an University, Xi'an, Shaanxi, China

## ABSTRACT

**Background:** With the growth of trajectory data, the large amount of data causes a lot of problems with storage, analysis, mining, *etc*. Most of the traditional trajectory data compression methods are focused on preserving spatial characteristic information and pay little attention to other temporal information on trajectory data, such as speed change points or stop points.

**Methods:** A data compression algorithm based on the spatio-temporal characteristics (CASC) of the trajectory data is proposed to solve this problem. This algorithm compresses trajectory data by taking the azimuth difference, velocity difference and time interval as parameters in order to preserve spatial-temporal characteristics. Microsoft's Geolife1.3 data set was used for a compression test to verify the validity of the algorithm. The compression results were compared with the traditional Douglas-Peucker (DP), Top-Down Time Ratio (TD-TR) and Opening Window (OPW) algorithms. Compression rate, the direction information of trajectory points, vertical synchronization distance, and algorithm type (online/offline) were used to evaluate the above algorithms.

**Results:** The experimental results show that with the same compression rate, the ability of the CASC to retain the forward direction trajectory is optimal, followed by TD-TR, DP, and then OPW. The velocity characteristics of the trajectories are also stably retained when the speed threshold value is not more than 100%. Unlike the DP and TD-TR algorithms, CASC is an online algorithm. Compared with OPW, which is also an online algorithm, CASC has better compression quality. The error distributions of the four algorithms have been compared, and CASC is the most stable algorithm. Taken together, CASC outperforms DP, TD-TR and OPW in trajectory compression.

## INTRODUCTION

Trajectory data has exploded with the popularity of ever-present positioning devices such as smart phones (*Liu et al., 2021*; *Han et al., 2017*). Trajectory data are also widely used for many purposes. The trajectory data of various vehicles can reflect the operational status of urban traffic, which can help traffic management departments to optimize road facilities and plan traffic lines (*Chen et al., 2020a*; *Makris et al., 2021a*; *Li et al., 2020*; *Chen et al., 2020b*). Pedestrian trajectory data can be used to analyze restaurants and tourist areas that

Corresponding author
Jinling Kong, jlkong@163.com

are of interest to pedestrians, as well as to assist shops in choosing business locations (*Jiagao et al., 2015*; *Ji et al., 2016*; *Kang et al., 2017*). At the same time, the huge amount of track data files can cause a lot of problems for data users. Taking common taxi trajectory data as an example, if trajectory data are collected every 2–3 s, a single car can generate about 15,000 trajectory points a day. With 67,000 taxis in Beijing, for example, 1 day's taxi trajectory data in Beijing takes up about 60 TB of memory (*Hao et al., 2019*). The sheer volume of data that is being transmitted and stored takes a huge amount of time, and the amount of time it takes to analyze that data is a frightening number. Trajectory data compression is an important approach to solving these problems (*Cao & Li, 2017*; *Lin et al., 2017*). As technology advances, the generation speed of trajectory data gets faster and faster, increasing the importance of trajectory data compression.

Trajectory data not only records the position of the object in motion, but also records time and speed information (*Chen, Xu & Franti, 2012*; *Wang, Gu & Ochieng, 2019*; *Yeh et al., 2018*; *Kontopoulos et al., 2020*). A large amount of valuable information is lost if the trajectory data is compressed only according to the geometric features of the trajectory, which is not conducive to later data analysis and data mining work. However, existing compression algorithms seldom pay attention to time characteristic information.

The Douglas-Peucker (DP) algorithm is a classical data compression algorithm that achieves compression by deleting redundant points according to the offset of track points (*Douglas & Peucker, 1973*). It has been widely used in many fields. Later, *Hershberger & Snoeyink (1992*, *1994)* improved the running speed of the DP algorithm for linear simplification. *Agarwal et al. (2005)* then improved the DP algorithm for the curve. However, these algorithms only focus on spatial features and have poor retention of the temporal features of the trajectory. Moreover, the value of the point deviation from the trajectory line can only be calculated after all trajectories are recorded. For this reason, *Meratnia & Rolf (2004)* proposed a time-proportional top-down Time Ratio (TD-TR) algorithm. It improved the classic DP algorithm by completely replacing the vertical distance with the synchronized Euclidean distance (SED), which is an algorithm that takes time characteristics into account. However, calculating the SED is a rather complex process, which results in a longer compression time than other algorithms, and it is still an off-line compression algorithm. Opening Window (OPW) is an online compression algorithm (*Keogh et al., 2001*). The OPW opens a sliding window at the track points, calculates in the window until the distance between the first and last points of the window is greater than the threshold, and then preserves these track points. So that the window is constantly updated to complete the online simplification of the track. The OPW algorithm solves the problem that the above algorithms has of needing all the data to make calculations, but it mainly considers the spatial characteristics in the compression, and does not fully consider the timing characteristics of the trajectory data. *Chen et al. (2020b)* proposed an algorithm, TrajCompressor compression, which uses the road matching principle to compress vehicle trajectory data. While the algorithm managed to achieve decent compression results, TrajCompressor can only compress vehicle trajectories and does not work for pedestrian paths. Moreover, a basic road map is required for road matching, which gives the algorithm narrow applicability as well as many preconditions.

In view of this, this article proposes a compression algorithm based on spatio-temporal characteristics (CASC). This algorithm takes into account both the spatial and temporal characteristics of the trajectory during compression. The CASC algorithm first judges the importance of each track point, then retains the important points and removes the minor points in order to better preserve the space-time characteristics of the track when the compression rate is the same. Three parameters are used to judge the importance of the trajectory point: azimuth difference, velocity difference and time interval. These three features correspond to turning point, speed change point and long time interval points in the trajectory, respectively.

The turning point is obviously very important for determining trajectory (*Makris et al., 2021b*; *Sousa, Boukerche & Loureiro, 2021*; *Zhan et al., 2014*). Because trajectory data compression is based on the premise that trajectory shape does not change, turning points should be preserved during compression (*Chen & Chen, 2021*; *Zhou, Qu & Toivonen, 2017*; *Fu & Lee, 2020*). The azimuth difference is used to measure whether a point is an turning point.

Trajectory data also has a temporal characteristic, in addition to the spatial characteristic (shape), which is the main characteristic that distinguishes trajectory data from general geometric data points (*Zhao et al., 2020*; *Leichsenring & Baldo, 2020*). The time characteristics of a trajectory are mainly reflected by its velocity, so the speed change point should also be preserved. The velocity difference is used to measure whether a point is an speed change point.

When the time interval between two adjacent track points is long, it is usually because no signal in the path has been recorded for a long time although the object is still in motion or the object stayed for a long time (*Bashir et al., 2022*; *Lin et al., 2021*). This is a relatively complex situation and can occur when the signal is lost because of the occlusion of buildings or mountains or when the moving objects move underground (*Kubicka et al., 2018*; *Koller et al., 2015*), even if the object is attracted to something and stops moving forward, these situations are worthy of attention for data mining, so these points should be preserved.

SASC actually considers whether each point in the trajectory belongs to these three types of points point by point. This algorithm is an online algorithm, which can complete compression without needing all data to be input. Compared with the above algorithms, the CASC algorithm is simple and efficient in calculating the importance of the points of the trajectory, with only O(n) time complexity. The CASC algorithm is applicable to all kinds of vehicles as well as pedestrian trajectory data and other types of trajectory compression, giving it a very wide applicability.

The rest of this article is organized as follows: the second section introduces the related work, mainly the ideas and processes of existing common compression algorithms; the third section mainly introduces how the algorithm calculates and determines the importance of each trajectory point, how to choose the turning point, speed change point, and long time interval point (stop point); the fourth section introduces the methods used to evaluate the results of data compression; the fifth section mainly introduces an experimental data compression situation and experimental methods; in the sixth section,

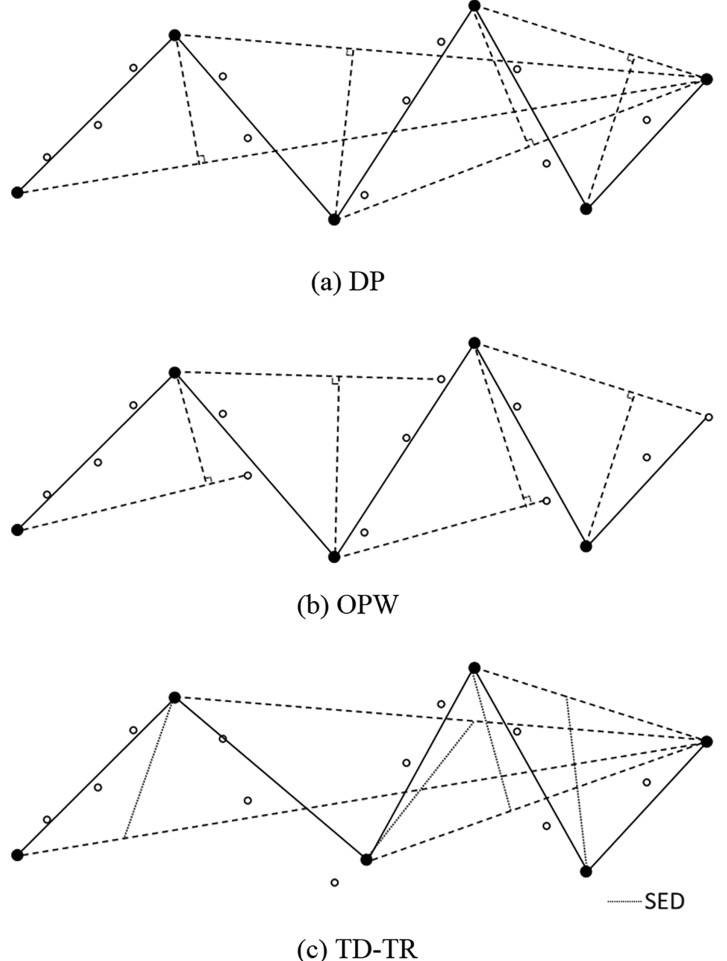

(a) DP

(b) OPW

(c) TD-TR

**Figure 1 (A) DP, (B) OPW and (C) TD-TR algorithm.**

the compression results are introduced and the compression algorithms are discussed and evaluated according to the results; and the seventh section gives the conclusion of the compression effect of the algorithm as well as possibilities for the future application of the algorithm.

## RELATED WORK

Three typical algorithms are selected from the above algorithms: the classic DP, the online algorithm OPW, and the TD-TR considering the time characteristics.

### DP

As shown in Fig. 1A, the DP algorithm defines the first point of the trajectory as the starting point and the last point as the floating point.

i) Calculate the vertical distance from all other trajectory points to the line connecting the starting point and the floating point, and find the point with the largest distance;

ii) If the maximum distance is greater than the given threshold, the original line segment is divided into two parts, and the maximum distance point becomes the front. The floating point of a line segment is also the starting point of the next line segment, and the maximum distance point is stored;

iii) Return to step i.

This cycle is repeated until all distances are less than the threshold, then the trajectory formed by the stored points is the compressed trajectory. The DP algorithm can only calculate the complete trajectory data (known start and end points) and is an offline algorithm.

## OPW

As shown in Fig. 1B, the OPW algorithm selects the first point in the trajectory as the starting point, the third point is set as a floating point, and both the starting point and the new starting point are stored.

i) Establish a connection between the starting point and the floating point. The points between the starting point and the floating point are regarded as the inspection points, and then the distance between the inspection points and this connecting line are calculated point by point.

ii) If all distances in step i are less than the given threshold, the next point of the floating point is regarded as a new floating point, and return to step i.

iii) If the distance of a certain point is greater than the threshold, then this point becomes a new starting point, and the third point after this point is used as a floating point, and returns to step i.

After the cycle is over, the trajectory formed by the stored points is the compressed trajectory. The OPW algorithm only needs to know the starting point of the trajectory and does not need to determine the end point of the trajectory. Therefore, the OPW algorithm can calculate the dynamically added data and is an online algorithm.

## TD-TR

The TD-TR algorithm (*Meratnia & Rolf, 2004*) is similar to the DP algorithm and can be considered as an improved version of the DP algorithm. TD-TR heuristically preserves both temporal and spatial information in compression. This is different from only considering spatial information in DP algorithm compression. The algorithm works like DP, but uses SED to replace the distance parameter in the DP algorithm, as shown in Fig. 1C.

TD-TR is an offline algorithm, which cannot be compressed in real time and needs to be compressed after all data is entered. SED is detailed in the Time characteristic evaluation subsection in the Compression Quality Evaluation section.

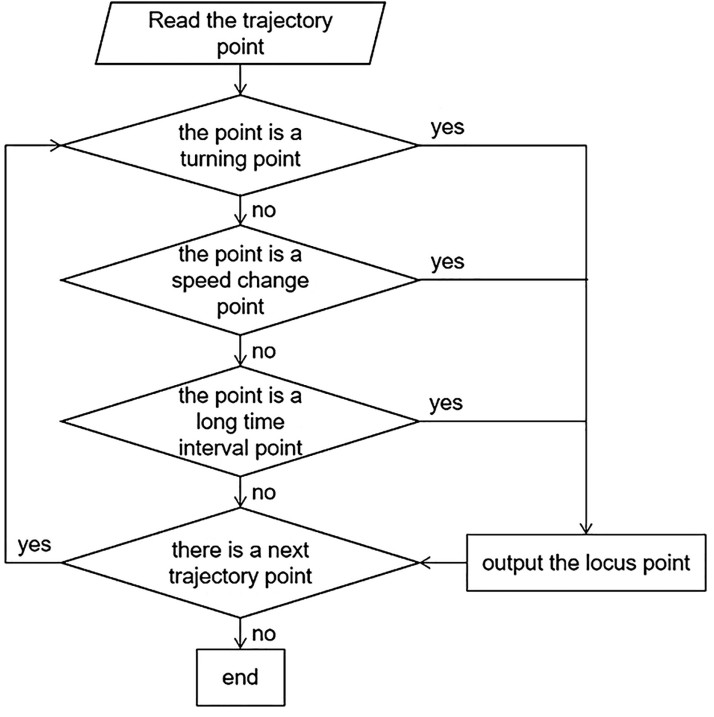

**Figure 2 Compression process.**

## THE CASC ALGORITHM

Trajectory data records the trajectory of the object in motion changing with time, which not only reflects the spatial position information of the object, but also describes its state of motion (*Kolesnikov & Franti, 2003*; *Nibali & He, 2015*). Common trajectory data is composed of continuous registration points including two-dimensional coordinate values and time stamps, such as $((x_1, y_1, t_1),(x_2, y_2, t_2)\ldots, (x_n, y_n, t_n))$. The two-dimensional coordinates of trajectory data are generally expressed using geographic coordinates (latitude and longitude) from GPS or BeiDou, *etc.* (*Long, Wong & Jagadish, 2013*). The purpose of CASC is to reduce redundancy points while preserving the original trajectory information as much as possible, thus achieving data compression. In this algorithm, the turning points, speed change points and long time interval points are used as the key data points. The compression process is shown in Fig. 2.

This flow chart shows that the main point of this algorithm is to judge whether each trajectory point is the turning point, the speed change point, or the long time interval point.

### Turning point

The difference of the azimuth angle of the track before and after a point can be used to determine whether that point is a turning point. Considering that geographic coordinates (latitude and longitude) are commonly used to describe location information in the trajectory data, azimuth calculations based on latitude and longitude need to be adopted.

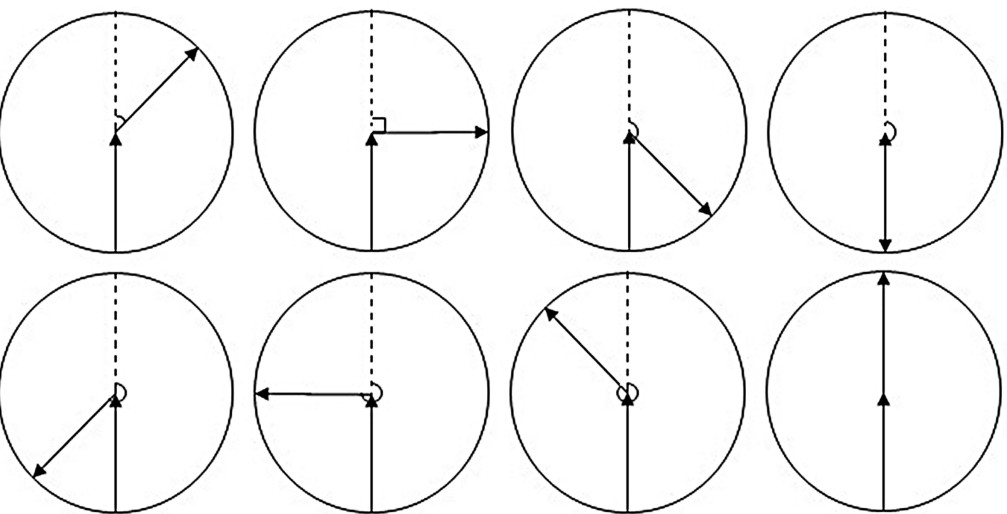

**Figure 3 Azimuth difference.**   

The azimuth of the directed line segment from the previous point to the current point, with $\alpha_i$ representing the azimuth from point $i-1$ to point $i$, can be calculated using the following formula. Assuming that the latitude and longitude of the previous point and this point are $long_{i-1}$, $long_i$, $lat_{i-1}$ and $lat_i$ ($i$ is the current point number and $i-1$ is its former point), then the azimuth of the forward direction of point $i$ is:

$$\alpha_i \begin{cases} A & when\ long_i \geq long_{i-1}\ and\ lat_i > lat_{i-1} \\ 360° + A & when\ long_i < long_{i-1}\ and\ lat_i > lat_{i-1} \\ 180° + A & when\ lat_i \leq lat_{i-1} \end{cases} \qquad (1)$$

$$A = \arctan\left(\frac{(long_i - long_{i-1}) \times \cos(lat_i)}{lat_i - lat_{i-1}}\right) \qquad (2)$$

When calculating the azimuth difference between the two trajectories, as shown in Fig. 3, when the azimuth difference is greater than 180°, the greater the difference, the smaller the degree of steering.

The following calculation is used to make the azimuth difference proportional to the degree of steering in order to determine the threshold value later:

$$\Delta\alpha_{i-1,i} = \begin{cases} |\alpha_i - \alpha_{i-1}| & when |\alpha_i - \alpha_{i-1}| \leq 180° \\ 360° - |\alpha_i - \alpha_{i-1}| & when |\alpha_i - \alpha_{i-1}| > 180° \end{cases} \qquad (3)$$

In the formula, $\Delta\alpha_{i-1,i}$ is the azimuth difference between point $i$ and $i-1$; $\alpha_{i-1}$, $\alpha_i$ is the azimuth of the $point_{i-1}$ and $point_i$. The greater the azimuth difference, the greater the degree of steering.

As shown in Fig. 4, the azimuth angle of each section of the trajectory is extracted and the azimuth difference ($\Delta\alpha_{2,3}$) is calculated. If it is small, it indicates that point 2 in Fig. 4 has a small degree of transition and has little influence on the shape of the trajectory after

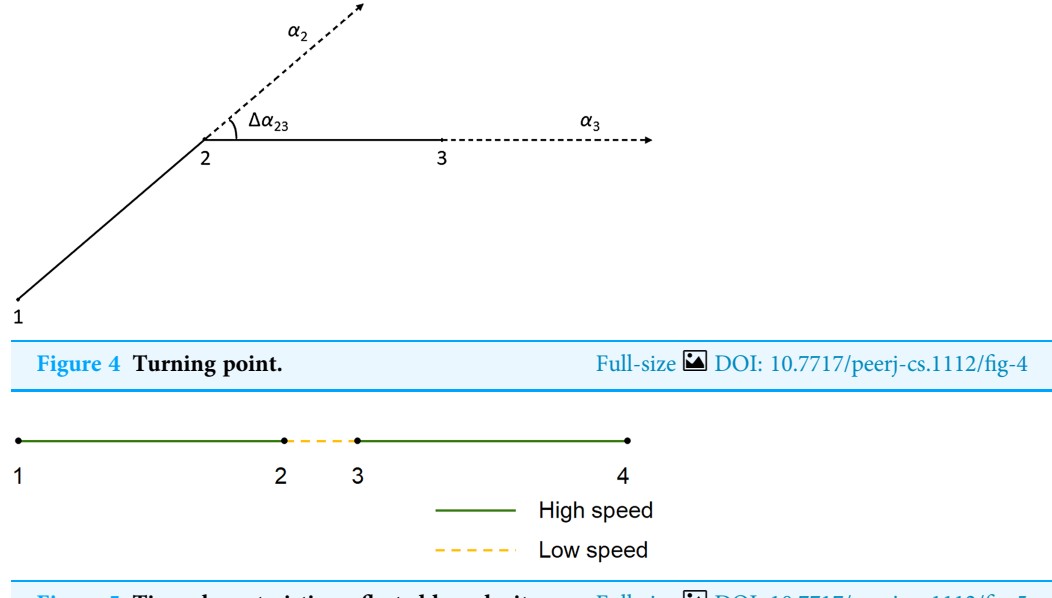

**Figure 4 Turning point.**

**Figure 5 Time characteristics reflected by velocity.**

deletion. Threshold $T_1$ is the value that we use to determine whether the azimuth difference is small or not.

## Speed change point

The time characteristics of a trajectory are mainly reflected by its velocity, as shown in Fig. 5: it is assumed that 1, 2, 3, and 4 are all on the same straight line and that the time intervals of each point are equal. If only the spatial characteristic is considered, since all points are on the straight line, both points 2 and 3 will be deleted during compression. However, point 2 and point 3 in the trajectory actually reflect low speed, which is an important point for trajectory data with time characteristics. Therefore, our algorithm uses velocity as a parameter to judge whether the trajectory point has strong time characteristics.

The speed based on geographical coordinates can be calculated as follows:

$$V_i = \frac{2R \times arcsin\sqrt{sin^2\left(\frac{lat_i - lat_{i-1}}{2}\right) + cos(lat_{i-1}) \times cos(lat_i) \times sin^2\left(\frac{long_i - long_{i-1}}{2}\right)}}{t_i - t_{i-1}} \quad (4)$$

where R is the radius of the earth, $long_{i-1}$ and $long_i$ respectively represent the longitude of $point_{i-1}$ and $point_i$, $lat_{i-1}$ and $lat_i$ respectively represent the latitude of $point_{i-1}$ and $point_i$, $t_{i-1}$ and $t_i$ respectively represent the moment of $point_{i-1}$ and $point_i$, and $v_i$ represents the mean velocity between $point_{i-1}$ and $point_i$. For convenience, the mean velocity of the line segment from $point_{i-1}$ to $point_i$ is abbreviated as the velocity at $point_i$ and denoted as $v_i$.

The calculation of the velocity difference between adjacent trajectory points is mainly used to preserve the speed change points in the trajectory points, where the velocity difference mainly consists of the absolute difference and relative difference. There are significant differences in the absolute speed difference between different travel modes: for example, the speed difference between cars at high speed and low speed is obviously

---

**Algorithm: CASC**

**INPUT**: Track data set point[long, lat,t,v], threshold $T_1$, threshold $T_2$

**OUTPUT**: The compressed trajectory data set

1. Initialization: i = 1

2. point$_0$ = [0,0,0,0]

3. **while** (point$_i$ ≠ Ø and point$_{i+1}$≠ Ø) {

4.     **do** $\Delta\alpha_{i,i+1}$, $\Delta v_{i,i+1}$, $\Delta t_{i-1,i}$, $\Delta t_{i,i+1}$ ;

5.     **If** ($\Delta\alpha_{i,i+1}$ <$T_1$ & $\Delta v_{i,i+1}$<$T_2$ & $\Delta t_{i-1,i}$<10 & $\Delta t_{i,i+1}$<10)

6.         i++;

7.     **Else**{print point$_i$,i++}

8. }

9. print point$_i$ //That means there is no point$_{i+1}$, point$_i$ is the last point in the trajectory can be output directly

---

greater than that between pedestrians at high speed and low speed. Considering this, the velocity difference mentioned in this algorithm refers to the relative difference of velocity rather than the absolute difference. The formula is as follows:

$$\Delta V_{i-1,i} = \frac{|V_i - V_{i-1}|}{V_{i-1}} \tag{5}$$

where $v_{i-1}$ and $v_i$ are the velocity values of point$_{i-1}$ and point$_i$. If there is a large difference in velocity between the track point and the point before and after, it indicates that this point is a speed change point and should be retained. The threshold for determining whether $\Delta v_{i-1,i}$ is large is $T_2$.

Both thresholds $T_1$ and $T_2$ are user definable.

## Long time interval points

Long time interval points are very important points in relation to the region of interest of the trajectory, or to preserve the start and end positions of the no-signal region. Considering the above reasons, the adjacent two points with a long time interval cannot be deleted.

$$\Delta t_{i-1,i} = t_i - t_{i-1} \tag{6}$$

$\Delta t_{i-1,i}$ is used to represent the time interval between two trajectory points. Based on the data set used in this algorithm, the time interval is generally 3–4 s. Therefore, when the experimental data set is compressed, any time interval between adjacent points that is longer than 10 s is preserved. The actual compression is determined according to the actual compression data set and compression requirements.

**Algorithm Flow.** Based on the above ideas, the pseudo-code of this algorithm is shown in Algorithm: CASC.

Because the first and last points of the trajectory represent start-stop information, they must not be deleted, and since the adjacent two points with a long time interval cannot be

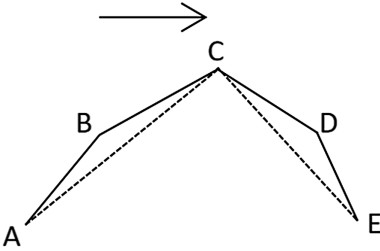

**Figure 6 Mean azimuth error.**               

deleted, both $\Delta t_{i-1,i}$ and $\Delta t_{i,i+1}$ are considered in the compression. The pseudo-code of the algorithm shows that the values of $T_1$ and $T_2$ directly affect the compression effect of the CASC compression algorithm. The size of the threshold is related to the compression rate as well as the loss of information. Generally, the higher the threshold, the higher the degree of compression, but with more loss of spatial and temporal characteristic information. Threshold size can be used to adjust these variables according to the purpose of compression and precision requirements in practical work.

# COMPRESSION QUALITY EVALUATION

## Compression rate

Compression rate is used as the common basic index to evaluate compression quality (*Leichsenring & Baldo, 2020*; *Lin et al., 2017*; *Zhao & Shi, 2019*). Compression rate is defined as the rate of the amount of data compressed to the amount of data before compression. The higher the compression rate, the better, but a higher compression rate also means more information is lost (*Liu et al., 2015*; *Kontopoulos, Makris & Tserpes, 2021*). Most of the compression algorithms mentioned above can achieve a compression rate of about 80% by changing the threshold (*Douglas & Peucker, 1973*; *Meratnia & Rolf, 2004*; *Keogh et al., 2001*; *Lv et al., 2015*). When the compression rate is the same, the degree of information loss is used to evaluate the advantages and disadvantages of different algorithms.

## Spatial characteristic evaluation

The main requirement of trajectory data compression is to maintain the forward direction of the trajectory before and after compression. The smaller the difference between the azimuth angle of the compressed trajectory segment and the azimuth angle before compression, the more complete the orientation information of the trajectory point is retained by compression. This can be evaluated using the mean azimuth error.

As shown in Fig. 6, the arrow is the forward direction, and point B is compressed during compression. The included angle between BC and AC is the azimuth error before and after point C compression. The azimuth error of each reservation point (uncompressed) is averaged, and the resulting value is the mean azimuth error, which is expressed as:

$$\overline{\Delta A} = \frac{\sum_{i=1}^{n} |\alpha_i - \alpha'_i|}{n} \tag{7}$$

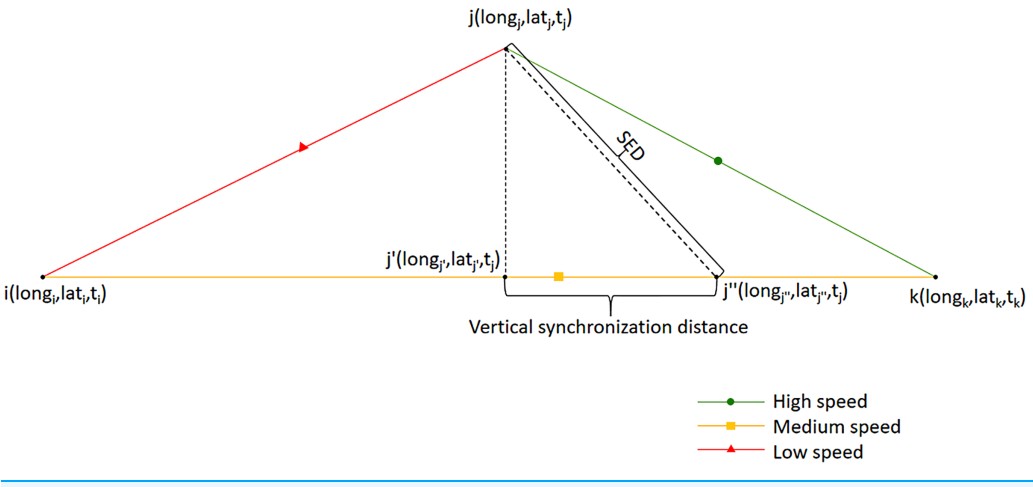

**Figure 7 Vertical synchronization distance.**

where $\overline{\Delta A}$ represents the mean azimuth error, $\alpha_i$ represents the azimuth of point i before compression, $\alpha_i{'}$ represents the azimuth of point i after compression, and n represents the total number of points after compression.

## Time characteristic evaluation

The time characteristics of the trajectory are mainly reflected in the form of velocity, where the vertical synchronization distance is used for evaluation, as shown in Fig. 7.

Since the locus point is compressed, the default state between the two locus points has uniform velocity. Figure 7 is divided into the trajectory before compression (upper half) and the trajectory after compression (lower half). Point j is the point before compression, point j′ is the projection of point j on the compressed trajectory, and point j″ is the position point at the same time as point j in the compressed trajectory. The distance between point j′ and j″ is the vertical synchronization distance before and after compression; jj″ is the SED used in TD-TR and other algorithms (*Meratnia & Rolf, 2004*). Point j′ is to evaluate the change of time characteristics of point j before and after compression, which will not be output in the compressed trajectories. When the algorithm calculates the coordinates of j′ and j″, it adopts the Mercator projection. The slope of the two-dimensional line segment formed by points i and k is calculated:

$$K = (\text{lat}_k - \text{lat}_i)/(\text{long}_k - \text{long}_i) \tag{8}$$

Constant b:

$$b = \frac{\text{long}_k \times \text{lat}_i - \text{long}_i \times \text{lat}_k}{\text{long}_k - \text{long}_i} \tag{9}$$

From the projection formula of point to line segment, the coordinates of j′ can be calculated:

$$long_{j'} = \frac{K(lat_j - b) + long_j}{K^2 + 1} \tag{10}$$

$$lat_{j'} = K \times long_{j'} + b \tag{11}$$

In proportion, the coordinates of j″ can be calculated:

$$long_{j''} = long_i + \frac{t_j - t_i}{t_k - t_i}(long_k - long_i) \tag{12}$$

$$lat_{j''} = lat_i + \frac{t_j - t_i}{t_k - t_i}(lat_k - lat_i) \tag{13}$$

From the coordinates of j′ and j″, we can get the vertical synchronization distance of point j before and after compression using the following equation:

$$d_{j',j''} = 2R \times arcsin\sqrt{sin^2\left(\frac{lat_{j'-}}{lat_{j''}}2\right) + cos(lat_{j''}) \times cos(lat_{j'}) \times sin^2\left(\frac{long_{j'-}}{long_{j''}}2\right)} \tag{14}$$

where R is the radius of the earth. The mean vertical synchronization distance can be obtained by averaging the vertical synchronization distance of all track points before and after compression, which can then be used to evaluate the time characteristic retention of the track after compression.

## EXPERIMENTS

### Datasets

In order to verify the effectiveness of this compression algorithm, Microsoft's Geolife1.3 data-set was used for the compression test (*Zheng et al., 2009*). This data-set contains 18,670 tracks, with a total length of 14,304 km, a total time of 12,953 h, and 24,876,978 total data points. The trajectory data includes both the trajectory data of pedestrians and vehicles (bike, car, bus, train, *etc*). Its trajectory contains various types of data, so there are already some trajectory compression algorithms that use it as the only data source (*Leichsenring & Baldo, 2020*; *Hao et al., 2019*).

Some of the data was eliminated because of either the small number of points recorded in part of the track (less than 100 track points) or the existence of a large range of trajectory point drift. The remaining data after deletion included 12,076 tracks and 20,657,826 data points.

### Experimental scheme

The data set was used to calculate each parameter according to the flow of the CASC algorithm, and compress it and then evaluate it according to above evaluation method. The DP, TD-TR and OPW algorithms were used to compress the experimental data for comparison.

## RESULTS

Table 1 shows that the time complexity of the CASC algorithm is O(n), indicating that the time efficiency of this algorithm is better than the other three algorithms. In the CASC, only three types of features and thresholds are compared point by point, so its time complexity is O(n). In DP and TD-TR, an auxiliary line needs to be generated before

**Table 1 Basic information of each algorithm.**

| Algorithm | Online/offline | Time complexity | Parameter |
|-----------|----------------|-----------------|-----------|
| CASC | online | $O(n)$ | speed, azimuth, time interval |
| DP | offline | $O(n^2)$ | distance |
| TD-TR | offline | $O(n^2)$ | SED |
| OPW | online | $O(n^2)$ | distance |

comparing distance/SED. Then select the point with the farthest distance/SED from the auxiliary line to generate a new auxiliary line, until the distance between each point and auxiliary lines are lower than the threshold. For each point, the distance/SED needs to be updated once for each new auxiliary line, so the complexity is $O(n^2)$. In OPW, every time the window is updated, the distance from each point to the auxiliary line is updated once, so it is also $O(n^2)$.

Figure 8 shows a starting trajectory and the trajectory compressed by four algorithms. The dense trajectory points in the upper left corner of the trajectory show that the trajectory stays in the upper left corner for a long time. The restaurant symbol in the picture indicates that there is food and beverage service at that location, which may explain the reason for staying a long time. In pedestrian trajectory data mining, this situation is an example of one that includes the interests and hobbies of the pedestrians, which is important information that cannot be deleted. Of all the above algorithms, CASC retains the track most completely (25 points are compressed to 20 points), while DP retains the least amount (25 points are compressed to three points). The right part of the track is the pedestrian moving along the same road for a long time, which is less important information. The CASC algorithm has the best compression effect (21 points are compressed to six points), while the TD-TR algorithm has the worst compression effect (21 points are compressed to 14 points). This compression strategy of CASC is based on the amount of information contained in the trajectory. Trajectories with small changes in velocity and direction have less information to mine, so we compress them considerably. Trajectories with large changes in speed and direction have a lot of latent information to be mined, so the compression level needs to be reduced to preserve the latent information. The DP algorithm also fails to compress the retrace path (in the middle of the track), and the distortion is too great compared with the original track. The other three algorithms keep the track well.

In order to analyze the influence of different thresholds, each algorithm uses different thresholds to test its performance. After statistics, the compression results of the algorithm are as follows.

As shown in the Table 2, CASC achieved good results in the compression of test data. The compression rate is mainly determined by the azimuth threshold $T_1$. The increases of velocity threshold $T_2$ also enhance the compression rate, but the effect is not as obvious as that of $T_1$. When the velocity threshold $T_2$ was 100%, the compression rate increased by 15.37% when the azimuth threshold rose from 8° to 16°, but the mean azimuth error and

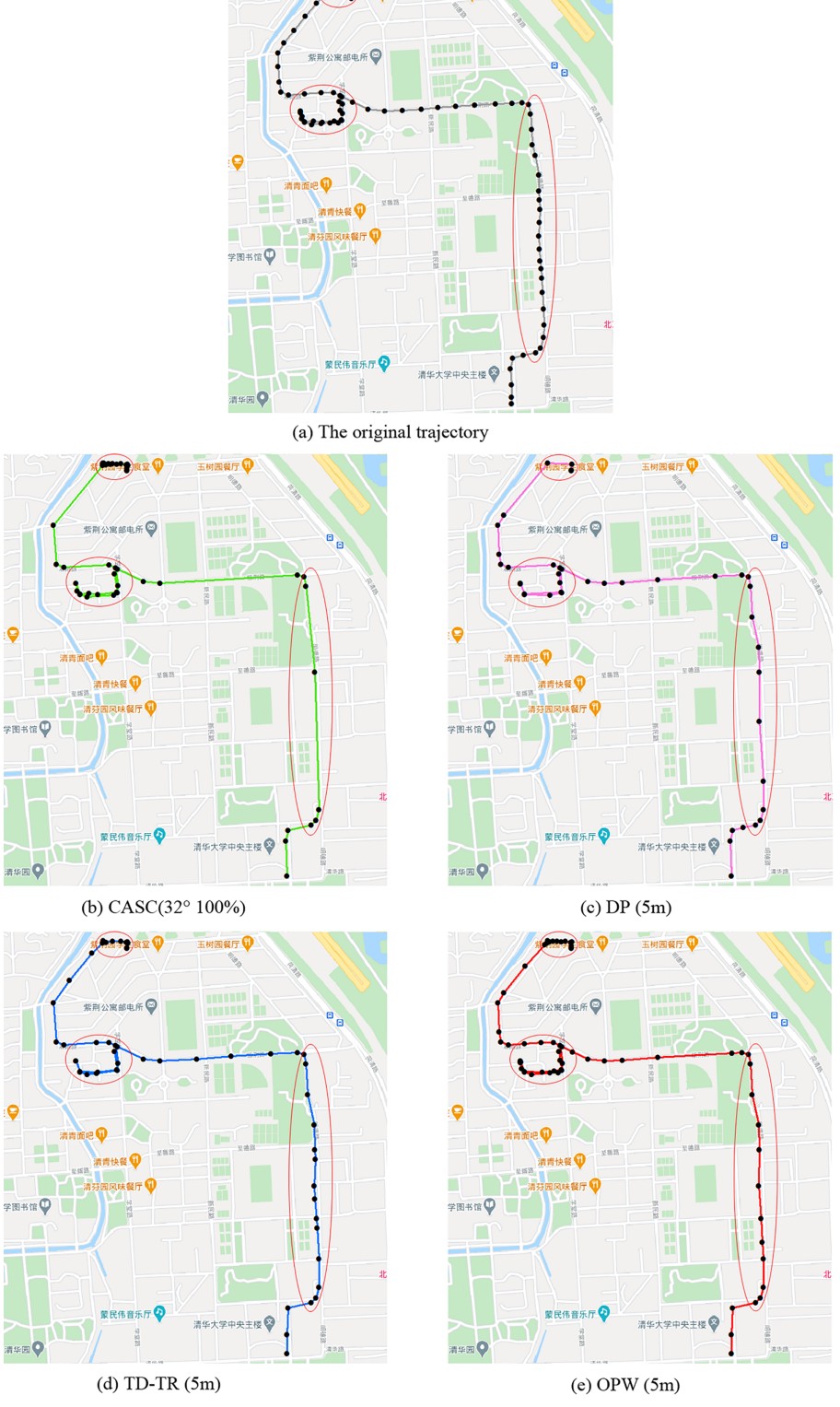

(a) The original trajectory

(b) CASC(32° 100%)

(c) DP (5m)

(d) TD-TR (5m)

(e) OPW (5m)

**Figure 8** **(A) Original trajectory, (B) CASC (32° 100%), (C) DP (5 m), (D) TD-TR (5 m), (E) OPW (5 m).**

**Table 2 Statistical table of CASC compression results.**

| Points | $T_1$ | $T_2$ | Points after compression | Compression rate/% | Mean azimuth error/° | Mean vertical synchronization distance/m |
|---|---|---|---|---|---|---|
| 20,657,826 | 8° | 50% | 12,564,089 | 39.18 | 1.8512 | 0.3345 |
| | | 100% | 11,599,369 | 43.85 | 1.8884 | 0.5568 |
| | | 200% | 9,965,335 | 51.76 | 1.9600 | 0.8666 |
| | 16° | 50% | 9,950,112 | 53.77 | 2.6018 | 0.7365 |
| | | 100% | 8,424,261 | 59.22 | 2.6296 | 1.1478 |
| | | 200% | 7,546,303 | 63.47 | 3.0611 | 1.8676 |
| | 32° | 50% | 5,003,325 | 75.78 | 7.9581 | 1.7765 |
| | | 100% | 4,164,617 | 79.84 | 8.9320 | 2.4654 |
| | | 200% | 3,435,396 | 83.37 | 10.7056 | 4.4353 |

the mean vertical synchronization distance only increased by 0.7412° and 0.5850 m, respectively. When the azimuth threshold rose from 16° to 32°, the compression rate increased by 20.62%, but the mean azimuth error and mean vertical synchronization distance increased by 6.3024° and 1.3176 m, respectively. This indicates that the distortion caused by compression when the azimuth threshold rose from 16° to 32° was higher than when it rose from 8° to 16°.

When the azimuth threshold was 16° and the velocity threshold rose from 50% to 100%, the compression rate increased by 5.45%, but the mean azimuth error and the mean vertical synchronization distance increased by 0.0278° and 0.4113 m. When the velocity threshold rose from 100% to 200%, the compression rate increased by 4.25%, but the mean azimuth error and mean vertical synchronization distance increased by 0.4315° and 0.7198 m, respectively. This indicates that the increase of the speed threshold from 100% to 200% resulted in greater distortion and a higher compression rate.

The compression results of DP, TD-TR and OPW are as follows.

In Table 3, D represents the distance in the DP and OPW algorithm, which can be understood as the distance of jj′ in Fig. 7. S represents the SED in the TD-TR algorithm, which can be understood as the distance of jj″ in Fig. 7.

The compression results of the four algorithms are respectively used to draw the line graph with the mean azimuth error and mean vertical synchronization distance as the vertical axis. As shown in Figs. 9 and 10, vertically, the lower the broken line, the smaller the error will be when the compression rate is the same. From the horizontal point of view, the right side of the line is better, which means that the algorithm can obtain a better compression rate under the same error.

Tables 2 through 3 show that the higher the compression rate of each algorithm, the greater both the mean azimuth error and the mean vertical synchronization distance are, and the more serious the distortion of the trajectory data, indicating that the compression rate is positively correlated with the error, all algorithms have the same conclusion.

The mean azimuth error is used to measure the retention of the direction information of the trajectory point. Figure 9 shows that the CASC algorithm is significantly superior to

**Table 3 DP, TD-TR, and OPW compression results.**

**DP**

| Points | D/m | Points after compression | Compression rate/% | Mean azimuth error/° | Mean vertical synchronization distance/m |
|---|---|---|---|---|---|
| 20,657,826 | 1 | 12,229,433 | 40.80 | 2.6342 | 0.2482 |
| | 3 | 7,319,068 | 64.57 | 7.2967 | 1.1673 |
| | 5 | 5,342,114 | 74.14 | 10.1105 | 1.9629 |
| | 10 | 3,433,331 | 83.38 | 15.5804 | 4.6648 |
| | 20 | 2,133,954 | 89.67 | 17.9638 | 9.3254 |

**TD-TR**

| Points | S/m | Points after compression | Compression rate/% | Mean azimuth error/° | Mean vertical synchronization distance/m |
|---|---|---|---|---|---|
| 20,657,826 | 1 | 12,010,460 | 41.86 | 2.4336 | 0.1962 |
| | 3 | 7,081,503 | 65.72 | 6.9825 | 0.8263 |
| | 5 | 5,315,259 | 74.27 | 10.0236 | 1.4438 |
| | 10 | 3,410,607 | 83.49 | 15.6332 | 3.1356 |
| | 20 | 2,115,361 | 89.76 | 18.3235 | 6.2254 |

**OPW**

| Points | D/m | Points after compression | Compression rate/% | Mean azimuth error/° | Mean vertical synchronization distance/m |
|---|---|---|---|---|---|
| 20,657,826 | 1 | 15,007,910 | 27.35 | 2.5461 | 0.1029 |
| | 3 | 10,052,098 | 51.34 | 6.8967 | 0.5327 |
| | 5 | 7,129,016 | 62.49 | 9.6105 | 0.9629 |
| | 10 | 5,129,338 | 75.17 | 13.4804 | 2.2136 |
| | 20 | 3,235,015 | 84.34 | 17.3638 | 4.8254 |

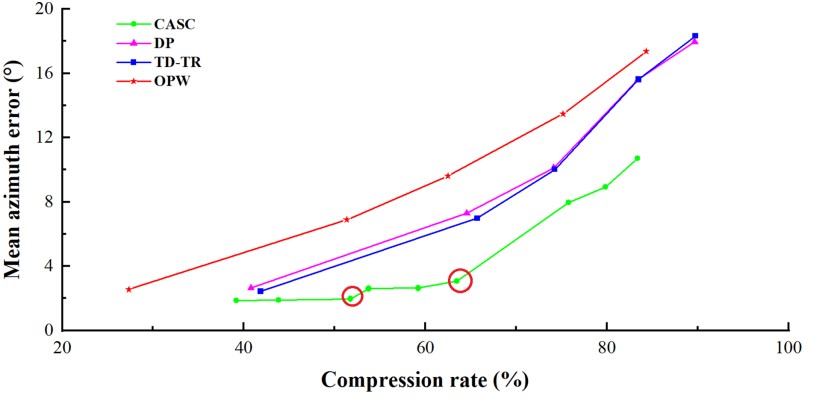

**Figure 9 Comparison of mean azimuth errors.**

other algorithms in its ability to retain this direction information. The results show that under the same compression rate, the CASC compression algorithm is able to best retain the orientation information of the trajectory data. When the compression rate is 50–70%, the mean azimuth error of CASC algorithm is reduced by more than 50% compared with

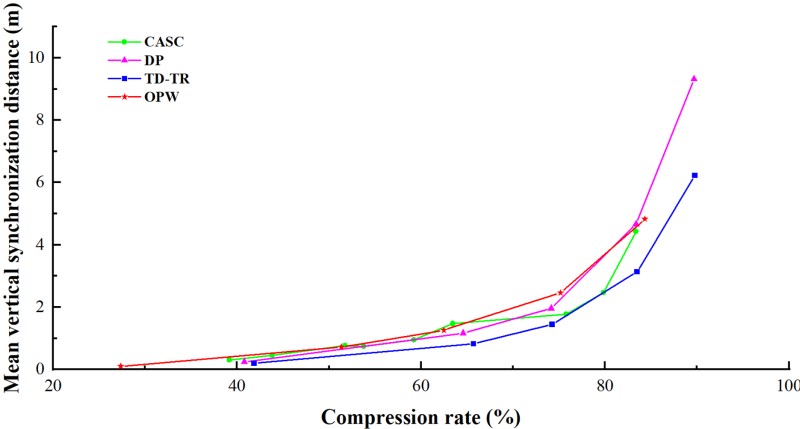

**Figure 10  Comparison of mean vertical synchronization distance.**

the other three algorithms. The mean azimuth error of CASC in Fig. 9 has two inflection points. Table 2 shows that these two inflection points correspond to two increases in threshold $T_1$ (8° to 16°, 16° to 32°), indicating that the appropriate selection of the azimuth threshold is the key to retaining forward direction information.

Mean vertical synchronization distance mainly reflects the algorithm's ability to maintain the time characteristics of the trajectory. Figure 10 and Table 2 show that CASC's vertical synchronization distance is closely related to the velocity threshold $T_2$. Under the same azimuth threshold $T_1$, the greater the velocity threshold, the greater the error will be. Even in the special case where the error decreases after the compression rate rises greatly, such as: when $T_1 = 32°$ and $T_2 = 50\%$, the mean vertical synchronization distance is less than when $T_1 = 16°$ and $T_2 = 200\%$. This conclusion can be drawn from both Fig. 10 and Table 2. These show that when the speed threshold value of this algorithm is 200%, the algorithm is unable to retain time characteristics, while when the value is 50–100%, it is better able to retain time characteristics. Therefore, the speed threshold value of this algorithm should not be too high. In addition, CASC still has a great advantage in comparing the vertical synchronization distance with OPW, which is also an online algorithm.

Two boxplots are drawn to analyze the error distribution of each algorithm under different thresholds, as shown in Figs. 11 and 12. Figure 11 shows the distribution of the azimuth error, and Fig. 12 shows the distribution of the vertical synchronization distance.

As shown in Fig. 11, when the azimuth threshold $T_1$ of CASC is constant, the change of its $T_2$ has little effect on the distribution of the azimuth error, which is distributed in a similar area. It shows that $T_1$ plays a key role in the azimuth error. However, DP, TD-TR, and OPW all use a single threshold (distance/SED), and the error increases with the increase of the threshold. With the increase of the threshold value, the distribution of azimuth error is also more scattered. The stability of the four algorithms is similar, and the

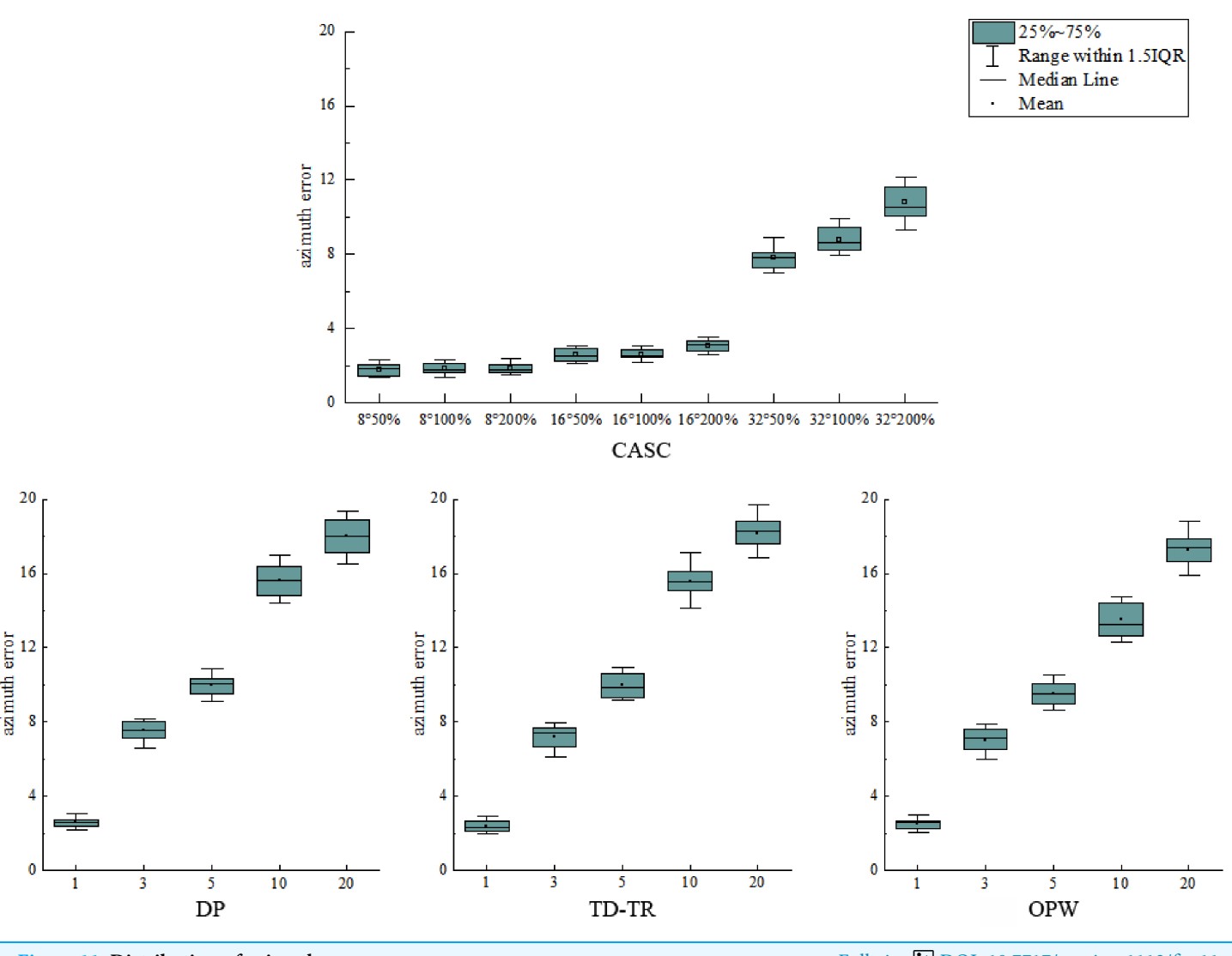

**Figure 11 Distribution of azimuth error.**

CASC average is the lowest. As shown in Fig. 12, the vertical synchronization distance of CASC is closely related to the velocity threshold $T_2$. Under the same azimuth angle threshold $T_1$, the larger the velocity threshold $T_2$, the larger the vertical synchronization distance. The dispersion degree of vertical synchronization distance distribution of CASC is obviously lower than that of DP, TD-TR and OPW. It shows that the CASC algorithm has better stability for temporal feature retention.

In general, the higher the threshold, the higher the compression rate and the greater the corresponding error. As shown in Fig. 9, compared with other algorithms, when the speed threshold value is about 100%, time characteristics can be better retained. Based on the above chart, the recommended threshold is $T_1 = 32°$ and $T_2 = 100\%$. Because the experimental data is GPS data, these suggested values are for GPS trajectories.

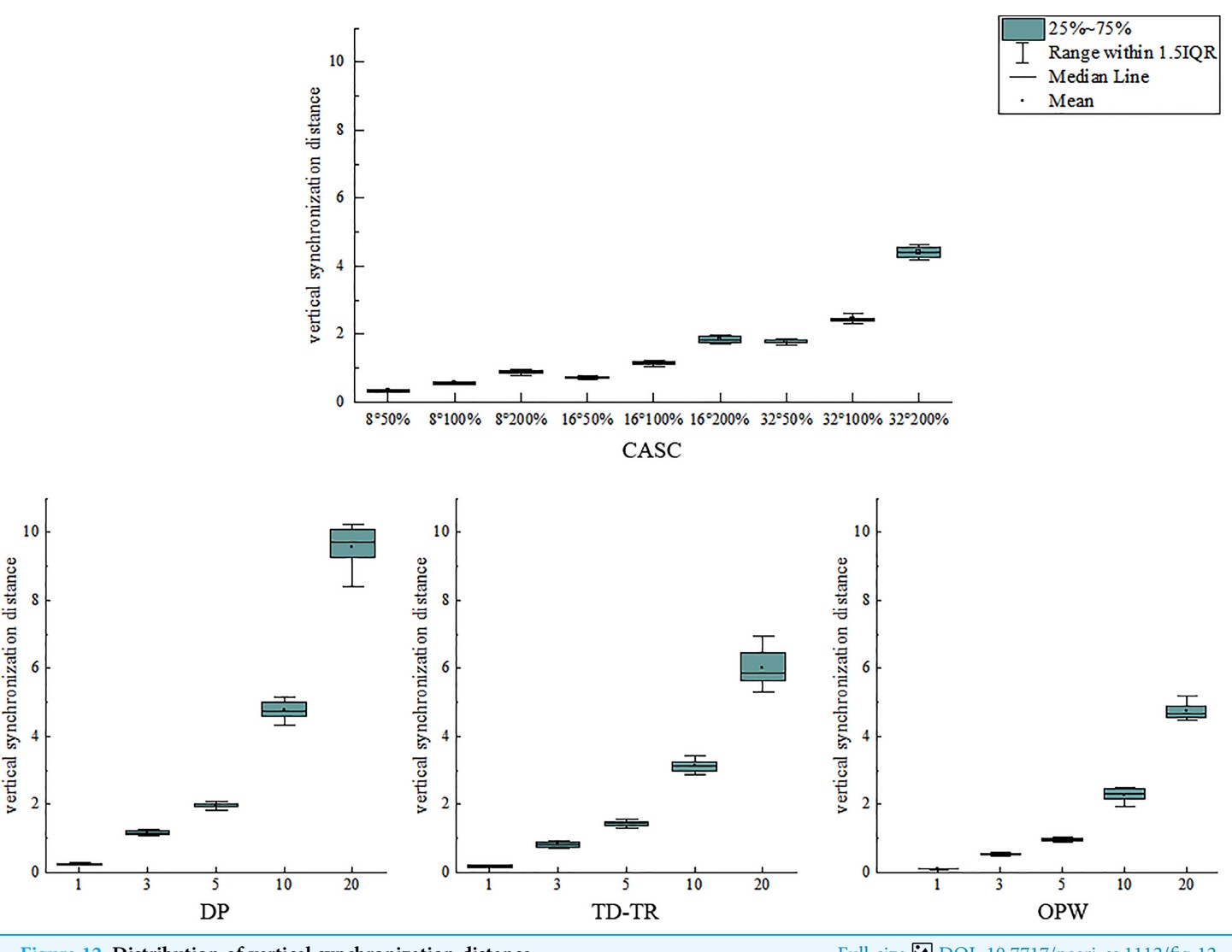

**Figure 12 Distribution of vertical synchronization distance.**

## CONCLUSIONS

In order to compress trajectory data, this article proposes a trajectory data compression algorithm based on spatio-temporal characteristics (CASC) that preserves geometric and time-domain features. The algorithm takes the azimuth difference, the velocity difference and the time interval as its parameters, and compresses the trajectory data while preserving spatial and temporal characteristic information. This study uses the Microsoft Geolife1.3 data set as an example, takes the mean azimuth error and the mean vertical synchronization distance as compression evaluation indexes, and compares and analyzes the information loss degree of the DP, TD-TR, OPW algorithms as well as the CASC algorithm proposed in this article under different compression rates. The results show that the CASC algorithm can be well applied to trajectory data compression and has significant advantages in keeping the direction of trajectory points.

Compared with the OPW algorithm, the mean azimuth error and mean vertical synchronization distance are both better with the CASC algorithm, which indicates that this algorithm improves the accuracy of the existing online compression algorithm. However, compared with the off-line algorithm, using SED as the threshold, the error of the vertical synchronization distance is large when the speed threshold is large using CASC, which needs to be further improved. The time complexity of this algorithm is $O(n)$, but its compression quality is better than that of the three comparison algorithms with a time complexity of $O(n^2)$, which not only improves the data compression quality but also saves compression time. Comprehensively comparing the error distributions of the four algorithms, the error of CASC has the highest stability.

Future work will focus on automatic acquisition of thresholds. Since the experimental data is GPS trajectory data, the recommended thresholds also apply to GPS trajectories. But for other types of trajectories, the steering and shifting may be different from GPS. Therefore, if CASC can automatically match the optimal threshold according to different data types, it will make CASC more widely applicable.

### Funding
This work was supported by the Department of Science and Technology of the Shaanxi Province key research and development projects (2020ZDLSF06-07). The funders had no role in study design, data collection and analysis, decision to publish, or preparation of the manuscript.

### Grant Disclosures
The following grant information was disclosed by the authors:
Department of Science and Technology of Shaanxi Province: 2020ZDLSF06-07.

### Competing Interests
The authors declare that they have no competing interests.

### Author Contributions
- Yanling Zhong conceived and designed the experiments, performed the experiments, analyzed the data, performed the computation work, prepared figures and/or tables, authored or reviewed drafts of the article, and approved the final draft.
- Jinling Kong conceived and designed the experiments, analyzed the data, prepared figures and/or tables, authored or reviewed drafts of the article, and approved the final draft.
- Juqing Zhang conceived and designed the experiments, analyzed the data, prepared figures and/or tables, authored or reviewed drafts of the article, and approved the final draft.
- Yizhu Jiang performed the experiments, prepared figures and/or tables, authored or reviewed drafts of the article, and approved the final draft.

- Xiao Fan performed the experiments, performed the computation work, prepared figures and/or tables, authored or reviewed drafts of the article, and approved the final draft.
- Zhuoyue Wang performed the experiments, prepared figures and/or tables, authored or reviewed drafts of the article, and approved the final draft.

## Data Availability

The Geolife1.3 data-set is available at Microsoft: https://www.microsoft.com/en-us/download/details.aspx?id=52367.

The CASC code is available in the Supplemental File.

## Supplemental Information

Supplemental information for this article can be found online at http://dx.doi.org/10.7717/peerj-cs.1112#supplemental-information.

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
