# Peer review of "A trajectory data compression algorithm based on spatio-temporal characteristics"

_PeerJ Computer Science, doi:10.7717/peerj-cs.1112_

## Round 0.1 · original submission · Major Revisions

I have reached a decision regarding your submission to PeerJ Computer Science, "A trajectory data compression algorithm based on spatio-temporal characteristics". The reviewers recommend Major Revisions, and they have raised a few issues in their reviews. Thus, I would like to ask you to address these issues carefully and revise your manuscript accordingly.

·

Basic reporting

1) Positive point: proposal of a raw (spatiotemporal) trajectory data compression strategy based on three trajectory features (azimuth difference, velocity difference and time interval);
2) To be improved/suggestions: why azimuth difference, velocity difference and time interval as input parameters to compress trajectory data? The paper misses background and related work sections presenting and discussing approaches for trajectory data compression, their strengths and limitations, as well as the novelties introduced by the author’s proposal in order to justify the choice for these three features and highlights the paper contributions;
3) To be improved/suggestions: the Related Work section of the paper describes the proposed compression method. A Related Work section typically presents a systematic literature review and compares the found related approaches. The paper must be well-structured according to my previous suggestion to define proper Background and Related Work sections. The current Related Work section could be titled “The CASC Algorithm” or something like that;
4) To be improved/suggestions: thresholds T1 and T2 are user defined? Isn’t there a way to define them automatically to let the compression method completely automatic? For time interval threshold, it is considered 10 seconds. It can be good for some datasets but not good for others. Isn’t there a way to compute it automatically based on the trajectory data point times?
5) To be improved/suggestions: subsection Time characteristic evaluation (line 291) details how the time characteristics of the trajectory are compressed. According to the explanation, it seems that a new point is generated (j’) in the compressed trajectory. Why a new point needs to be generated? Why not preserve only the original trajectory points where we have speed changes, as shown in Figure 1? Please clarify this;
6) To be improved/suggestions: Conclusions section does not discuss future works. What else can be done to improve CASC compression quality, e.g., to improve the compression rate? Some of the other algorithms obtained better compression rates when compared to CASC…

Experimental design

1) Positive point: a comparison of the proposed strategy is accomplished considering other popular trajectory data compression algorithms.

Validity of the findings

1) Positive point: experimental evaluation demonstrates an advantage of CASC regarding retention of the trajectory direction and vertical synchronization distance against other compression algorithms in the literature;
2) Positive point: an evaluation of proposed thresholds T1 and T2 is also accomplished in order to suggest the best value for them (T1 = 32 and T2 = 100);
3) To be improved/suggestions: why the CASC complexity is O(n) and the other algorithms execute in O(n2)? n is the number of trajectory points? Justify these complexities.

Additional comments

no comments.

·

Basic reporting

This paper proposes a compression algorithm based on spatio-temporal
characteristics of a trajectory during compression. The main contribution is an algorithm that analyzes the importance of each spatio-temporal point, keeps the crucial points and removes the minor points aiming at preserving the space-time characteristics of the track when the compression rate is the same.

The algorithm uses three parameters: azimuth difference, velocity difference, and time interval to judge the importance of the trajectory point. The proposed solution is an online algorithm that can complete compression without needing all data to be input.

The authors claim that the proposed algorithm is simple and efficient in calculating the importance of the trajectory points, with only O(n) time complexity. Experiments were conducted and the proposed approach outperforms the competitors (DP, TD-TR and OPW).

The manuscript is well written and organized. The problem is clearly defined, but the Related Work section presents the proposed approach instead discussing the state-of-the-art. Thus, the authors must include a deeper discussion about competitors and position their work according to them.

Experimental design

There are many flaws in the experimental design of this paper. First, the experiments used only one dataset, which is insufficient to show that the proposed solution is valid. For example, Figure 7 shows only one compress trajectory, and it is not evident that the algorithm will perform well in all cases.

Besides, Tables 2, 3, 4 e 5 are not conclusive. The use of mean (i.e., mean azimuth error and mean vertical synchronization distance) as a validation metric for the proposed approach performance is highly doubtful. Indeed, we need to understand the distribution of such metrics to understand how good is the proposed model.

Validity of the findings

The submission clearly defines the research question, relevance, and proposed solution. However, the authors did not identify the knowledge gap existing within the literature where this solution might fit in. In this sense, a deeper analysis of the literature must be done, allowing o a more comprehensive understanding of the validity of the contributions. Consequently, the scientific contributions are not convincing at all.

The main limitation of the proposed approach is to be dependent on two parameters that are pretty sensitive to trajectories data skew. Suppose that we have many trajectories with different shapes in a trajectory dataset, some of which have high deviations and others with slight variations. In this case, the choice of the parameters is not evident and straightforward, which may generate different results.

---

## Round 0.2 · accepted · Accept

After reading this new version of the article and based on the second round revision comments, I can state that the authors addressed the previous comments properly. This article can be accepted.

·

Basic reporting

The authors agree and consider all the suggestions for paper improvement w.r.t.: (i) better paper motivation and literature discussion; (ii) a Related Work section; (iii) justifications for some design decisions of the proposed approach; and (iv) future works. I think the paper is now acceptable.

Experimental design

No comments.

Validity of the findings

The authors better discuss the time complexity of the compared algorithms, which was missing in the previous paper version.

Additional comments

No more comments.